

# A simple mechanistic model of the invasive species *Heracleum sosnowskyi* propagule dispersal by wind

Ivan Chadin[1], Igor Dalke[2], Denis Tishin[3], Ilya Zakhozhiy[2] and Ruslan Malyshev[2]

[1] Molecular Biology Facility, Institute of Biology of Komi Science Centre of Ural Branch of Russian Academy of Sciences, Syktyvkar, Komi Republic, Russian Federation
[2] Laboratory of Plant Ecological Physiology, Institute of Biology of Komi Science Centre of Ural Branch of Russian Academy of Sciences, Syktyvkar, Komi Republic, Russia
[3] Institute of Environmental Sciences, Kazan Federal University, Kazan, Republic of Tatarstan, Russian Federation

## ABSTRACT

**Background**. Invasive species are one of the key elements of human-mediated ecosystem degradation and ecosystem services impairment worldwide. Dispersal of propagules is the first stage of plant species spread and strongly influences the dynamics of biological invasion. Therefore, distance prediction for invasive species spread is critical for invasion management. *Heracleum sosnowskyi* is one of the most dangerous invasive species with wind-dispersed propagules (seeds) across Eastern Europe. This study developed a simple mechanistic model for *H. sosnowskyi* propagule dispersal and their distances with an accuracy comparable to that of empirical measurements.
**Methods**. We measured and compared the propagule traits (terminal velocity, mass, area, and wing loading) and release height for *H. sosnowskyi* populations from two geographically distant regions of European Russia. We tested two simple mechanistic models: a ballistic model and a wind gradient model using identical artificial propagules. The artificial propagules were made of colored paper with a mass, area, wing loading, and terminal velocity close to those of natural *H. sosnowskyi* mericarps.
**Results**. The wind gradient model produced the best results. The first calculations of maximum possible propagule transfer distance by wind using the model and data from weather stations showed that the role of wind as a vector of long-distance dispersal for invasive *Heracleum* species was strongly underestimated. The published dataset with *H. sosnowskyi* propagule traits and release heights allows for modeling of the propagules' dispersal distances by wind at any geographical point within their entire invasion range using data from the closest weather stations. The proposed simple model for the prediction of *H. sosnowskyi* propagule dispersal by wind may be included in planning processes for managing invasion of this species.

Corresponding author
Ivan Chadin, chadin@ib.komisc.ru

## INTRODUCTION

Invasive species are one of the key elements of human-mediated ecosystem degradation and ecosystem services impairment worldwide. Dispersal of propagules is the first stage

of introduction and the driving force behind biological invasion (*Williamson, 1996*; *Richardson et al., 2000*; *Nehrbass et al., 2007*). The success and rate of biological invasion directly depend on the mobility of a species, its ability to spread over long distances, and the effectiveness of the use of dispersal agents (*Pyšek & Richardson, 2007*; *van Kleunen et al., 2015*).

In recent decades, *Heracleum sosnowskyi* Manden. (Apiaceae), an invasive plant species, has attracted considerable attention. Its invasion has significant environmental and socio-economic impacts in Eastern Europe and the European part of Russia (*Satsyperova, 1984*; *Chadin et al., 2017*; *Ozerova & Krivosheina, 2018*; *Gudžinskas & Žalneraviius, 2018*). A significant part of its invasion range lies between 48.6°N in the South and 72.6°N in the North, and it also occupies territories between 15.0°E on the West and 69.5°E in the East. *H. sosnowskyi* plants do not have any vegetative reproduction structures. Consequently, the invasive success of this species depends directly on the number of propagules and their ability to disperse (*Dalke et al., 2015*; *Gudžinskas & Žalneraviius, 2018*).

There have been several attempts to assess wind dispersal distances of *Heracleum* species mericarps. Researchers have conducted experiments in wind tunnels (*Clegg & Grace, 1974*), analyzed seedling density dependence on distance from the maternal plant (*Pergl et al., 2011*), analyzed aerial photographs (*Müllerová et al., 2005*; *Moravcova et al., 2007*), and directly measured propagule flight distance under field conditions (*Jongejans, Skarpaas & Shea, 2008*; *Wojewódzka et al., 2019*).

There is consensus in the existing literature that most propagules of *Heracleum mantegazzianum* Sommier & Levier, which is an invasive species that is phylogenetically and eco-physiologically close to *H. sosnowskyi*, fall no further than 5 m from maternal plants. Dispersal over distances > 10 m should be considered as long-distance dispersal (LDD), even for the tallest species of this genus (*Pergl et al., 2011*). Long-distance dispersal makes a major contribution to the spread of plant species (*Nathan & Muller-Landau, 2000*). The average rate of linear spread found for *H. mantegazzianum* in the Czech Republic was assessed as 10.8 m/year, with a maximum value of 26.7 m/year (*Müllerová et al., 2005*). Several authors have reported that streams and human activity are likely to play a major role in LDD of *H. mantegazzianum* propagules. Streams may transfer *H. mantegazzianum* propagules up to hundreds of meters, whereas most propagules dispersed by water are transported over shorter(<40 m) distances (*Trottier, Groeneveld & Lavoie, 2017*).

Despite the LDD potential of water, wind is often the only dispersal agent at many sites occupied by invasive *Heracleum* species, and we need to re-assess wind as a vector for LDD of these species. We expect that propagule transfer distances by wind may be strongly underestimated because of difficulties in the registration of LDD events. The propagules of Apiaceae are adapted precisely for anemochoria. The oval-shaped mericarps of *Heracleum* species are flattened with a wing-like border, and these propagules are catapulted from mature umbels when the dry and elastic stems are pushed by gusts of wind or moving objects (*Levina, 1957*; *Tackenberg, 2003*; *Vittoz & Engler, 2007*). We found two estimates in the literature of LDD by wind for *Heracleum* species: up to 100 m for *H. mantegazzianum* (*Ochsmann, 2008*) and up to 50 m for *H. sosnowskyi* (*Kondratiev, Budarin & Larikova, 2015*), although these reports were not confirmed by any instrumental
measurements. It is known that wind is able to transport *H. sosnowskyi* propagules for distances up to hundreds of meters by dragging them along on the flat surface of icy roads (*Krivosheina, Ozerova & Petrosyan, 2020*) or similar surfaces; however, modeling of this specific process was beyond the scope of the present study.

Anemochorous seed dispersal is well studied for many plant species and has been generalized in a number of mechanistic models of varying complexity, as reviewed by *Nathan et al. (2011)*. However, to date, none of these models have been used to describe seed dispersal in *Heracleum* species.

The aim of this study was to develop a simple mechanistic model that enables determination of the distance of *H. sosnowskyi* propagule dispersal by wind during the period from fruit ripening until the formation of snow cover, with an accuracy comparable to that of empirical measurements. Such a model should enable assessment of the dispersal kernel as well as possible LDD events for different parts of the *H. sosnowskyi* invasion range, where wind is the only available dispersal agent. In addition, this model should aid in reassessing the significance of wind as an LDD agent for *H. sosnowskyi*, and provide practical recommendations for invasion management using weather station data alone and a mechanistic description of one of the factors determining *H. sosnowskyi* invasiveness.

## METHODS

### Model Development

The distance of the horizontal flight of propagules in the airflow is known to depend mainly on three parameters: terminal velocity, release height of the propagules, and mean horizontal wind speed (*Levin et al., 2003*; *Dauer, Mortensen & Humston, 2006*; *Jongejans, Skarpaas & Shea, 2008*; *Nathan et al., 2011*). The simplest mechanistic model of seed dispersal by wind is a simple ballistic model suggested by *Dingler (1889)* cited in *Nathan et al. (2011)* and formalized as an equation by Schmidt at the beginning of the 20th century (*Nathan et al., 2011*):

$$D = \frac{h_r \bar{u}}{V_t} \tag{1}$$

where $D$ is the distance of horizontal flight, $\bar{u}$ is the mean horizontal wind speed, $h_r$ is the seed release height, and $V_t$ is the terminal velocity (the constant velocity of a seed falling in still air).

This simple ballistic model assumes several simplifications: (1) the seed reaches terminal velocity immediately after release, (2) the horizontal speed of the seed is equal to the horizontal speed of the air flow, (3) the wind does not change speed in the vertical direction, (4) there is no turbulence, and (5) the air flow does not meet obstacles in its path.

Currently, there are a number of models that omit all these restrictions; however, this leads to a complication of the mathematical apparatus, for example, requiring the use of the Navier–Stokes equations (*Nathan et al., 2011*). We have presumed that it is possible to accept most of the restrictions of the simple ballistic model. Some of these restrictions may be accepted based on *H. sosnowskyi* characteristics and its typical habitats in invaded areas. Other restrictions may be accepted after experimental verification.
The assumption of rapid deceleration of propagules to terminal velocity can be empirically verified by measuring the time taken for a seed to fall from various heights. The assumption that there is no significant difference between the horizontal speeds of the wind and the propagules flying in the airflow can also be tested experimentally. Air turbulence significantly affects the horizontal flight distance only for propagules with a very low terminal velocity: $0.07 \leq V_t < 0.3$ m / s (*Nathan et al., 2011*), and depends on land surface heterogeneity. Our model only considers dispersal from a single plant in an open space to allow the assumption that there is no turbulence and that the airflow is not disrupted by obstacles. In relation to these assumptions, the *H. sosnowskyi* invasion range is mainly located on flatlands, and solitary generative *H. sosnowskyi* plants located at a distance of more than several dozen meters from monostands or any other tall vegetation ($> 3$–5 m) are not uncommon.

The vertical gradient of the horizontal wind speed may be incorporated into a simple ballistic model. One of the equations describes the wind gradient phenomena as:

$$v_z = v_g \left(\frac{z}{z_g}\right)^\alpha \qquad (2)$$

where $v_z$ is the speed of the wind at height $z$, $v_g$ is the speed of the wind at height $z_g$ and $\alpha$ is the Hellmann exponential coefficient, which represents the degree of surface roughness and air stability *Cleveland & Morris (2013)*. Equation (2) can be rewritten with the notation used for the simple ballistic model (Eq. (1)):

$$v_h = v_{hr} \left(\frac{h}{h_r}\right)^\alpha \qquad (3)$$

where $v_h$ is the speed of the wind at height $h$, $v_{hr}$ is the speed of the wind at height $h_r$ (the release height), and $\alpha$ is the Hellmann exponential coefficient.

Therefore, the mean horizontal wind speed ($\bar{u}$) in the simple ballistic model may be replaced by the continuously changing (decreasing) wind velocity $v_h$. If we accept the assumption about wind and seed velocity equivalence, then the horizontal distance of seed flight may be determined by a definite integral of the wind velocity change rate. The $h$ in Eq. (3) depends on time:

$$h = v_t t_f - v_t t \qquad (4)$$

where $h$ is the height at time $t$, $t_f$ is the total time of seed fall from the release height to ground level, and $v_t$ is the seed terminal velocity. Next, the h in Eq. (3) is replaced with Eq. (4) to obtain the wind velocity change rate $v_h(t)$:

$$v_h(t) = v_{hr} \left(\frac{v_t t_f - v_t t}{h_r}\right)^\alpha \qquad (5)$$

where $v_{hr}$ –is the speed of the wind at height $h_r$ (the release height), $h$ is the height at time $t$, $t_f$ is the total time of seed fall from the release height to the ground level, $v_t$ is the seed terminal velocity, and $\alpha$ is the Hellmann exponential coefficient. The integration of Eq. (5) allows us to determine the horizontal flight distance of the seed:

$$D = \int_0^{t_f} v_{hr} \left(\frac{v_t t_f - v_t t}{h_r}\right)^\alpha \qquad (6)$$

Here after, the model represented by Eq. (6) is referred to as the gradient model. To test whether the gradient model sufficiently described seed dispersal in *H. sosnowskyi*, we collected empirical data from seed release experiments.

## Data collection

The characteristics of propagules and release heights were determined for *H. sosnowskyi* plants at two geographically distant sites located in two regions in Russia. Samples were collected in the suburbs of Syktyvkar City (Komi Republic, Russia) in 2018 (the North Group), and in the vicinity of Kazan City (Republic of Tatarstan, Russia) in 2017–2018 (the South Group). The North Group plants were from two sites with coordinates: 61.65°N, 50.74°E and 61.70°N, 50.80°E. The South Group plants were from two sites with coordinates: 55.80°N, 49.16°E and 55.94°N, 49.27°E. The mericarps were collected, and measurements made in typical monostands of the species. The heights of the central and lateral umbels were measured as the distance from the root-stem junction to the top of the main or lateral shoots of the plant.

The propagules were randomly selected from the bulk samples collected in the field. The air-dry weight of propagules was determined using analytical balances with an accuracy of 0.0001 g. To measure the surface area of the propagules (the area of one side), images of the propagules were obtained using a flatbed scanner at a resolution of 600 dpi. The area of the propagule images was determined using ImageJ software (*Schneider, Rasband & Eliceiri, 2012*).

Wing loading ($WL$, g / cm$^2$) was calculated as the ratio of the propagule mass (g) to the area of one of its sides (cm$^2$). The propagule falling velocity ($V_t$, m / s) was determined by measuring the falling time ($t$, s) from different release heights ($h$, m). The measurements were made indoors in the absence of airflow at room temperature. Propagules were dropped from heights between 0.80 and 4.28 m. The moment of propagule landing on the floor was visually recorded. The falling time was measured using a stopwatch. Each propagule was dropped five times, and the median values were used for subsequent calculations. The North Group consisted of 70 propagules, and the South Group, 60 propagules.

It was not possible to find several hundred propagules of *H. sosnowskyi* with the same standard area and mass. Therefore, we used artificial models of *H. sosnowskyi* propagules made of paper, the density of which was close to the average density of propagules of this species in the air-dry state. The contours of the artificial propagules were drawn as ellipses with a major axis of 1.35 cm and a minor axis of 0.88 cm (Fig. 1B).

The traits of the artificial propagules (median values and interquartile range, $N = 36$) were as follows: mass, $21.5 \pm 0.5$ mg; area, $0.966 \pm 0.018$ cm$^2$; and wing loading, $0.0220 \pm 0.0004$ g/cm$^2$. The median terminal velocity measured from a release height of 4.15 m was $1.72 \pm 0.11$ m/s (N = 10). We conducted all field measurements using these artificial propagules. The main difference between natural and artificial propagules was their surface structure. The artificial propagules were much smoother, and this may have influenced their air drag force. Therefore, the difference between the horizontal wind velocity and propagule velocity may be slightly larger for artificial propagules, and the distances we observed may be slightly shorter for these propagules than for natural

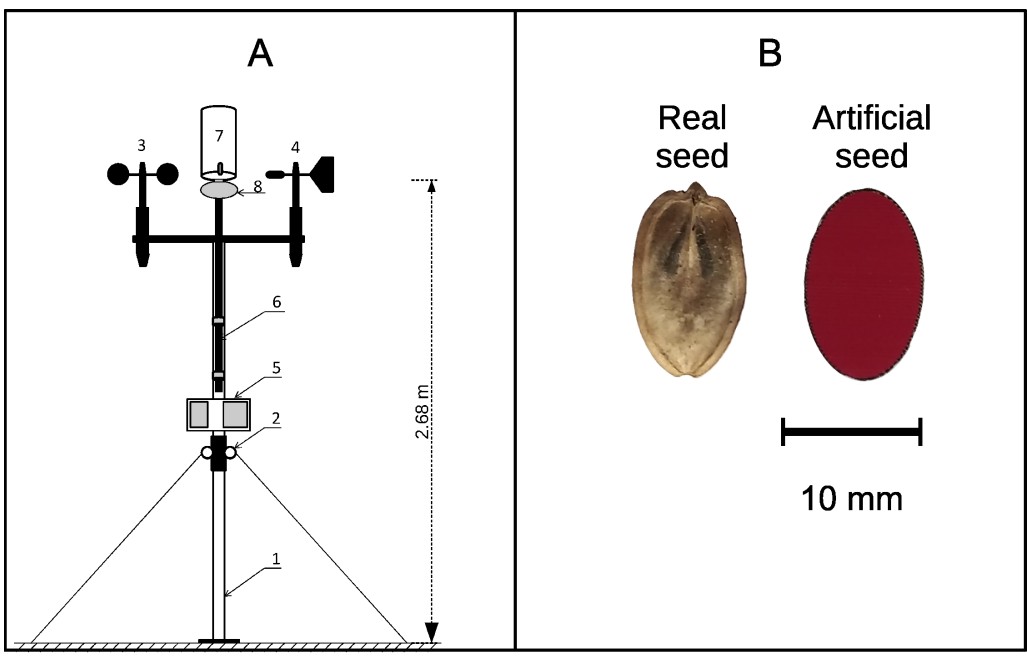

**Figure 1** (A) Diagram of the device used for propagule dispersal measurements; (B) a real Heracleum sosnowskyi propagule and an artificial propagule used for model testing. In (A): Two-section support rod, 2: rings for struts and securing the rod, 3: anemometer, 4: weather-vane, 5: data recording block, 6: removable rod for attaching the propagule-holding container, 7: propagule container, and 8: container cover with a lock for remote propagule release.

propagules of similar sizes. However, the main propagule trait that influences propagule dispersal distance by wind is the terminal velocity, and our artificial propagules reproduced the terminal velocity of *H. sosnowskyi* well. This good terminal velocity match was achieved using a form, size, and paper density close to that of real mericarps. We assumed that we captured the main traits of *H. sosnowskyi* propagules that are important for dispersal by wind. Artificial propagules were launched under natural conditions using a specially designed and manufactured device (Fig. 1A).

Measurements of horizontal flight distances of artificial propagules were carried out in batches of 10–20 propagules. A batch of propagules was loaded into the container and all propagules were dropped simultaneously. The wind speed was recorded by videotaping the anemometer readings. The horizontal flight distance of propagules was measured using a surveyor's tape with an accuracy of 0.01 m. Measurements were performed in Syktyvkar on 26th January, and 11th, 12th and 17th February, 2020 during daylight hours (11:00 h–16:00 h). As a result, 37 launches of batches of paper diaspores of *H. sosnowskyi* were performed at wind speeds from 0 to 9 m/s.

All data analyses were conducted in R (http://www.R-project.org). The primary data and R-script used for performing calculations are available at the Zenodo repository (https://doi.org/10.5281/zenodo.3837647).

**Table 1** Traits of *Heracleum sosnowskyi* propagules (North Group, $N = 70$).

| Summary results | Mass, mg | Area, cm$^2$ | Wing loading, g / cm$^2$ | Falling velocity [a], m / s |
|---|---|---|---|---|
| Mean | 11.0 | 0.57 | 0.019 | 1.59 |
| Median | 11.0 | 0.56 | 0.019 | 1.62 |
| Standard deviation | 4.3 | 0.12 | 0.006 | 0.25 |
| IQR | 3.2 | 0.17 | 0.005 | 0.27 |
| Standard error of mean | 0.5 | 0.01 | 0.001 | 0.03 |

Notes.
[a] The falling velocity was determined for a release height of 2.68 m.

# RESULTS

The sample sizes of the North Group ($N = 70$) and South Group ($N = 60$) of propagules were used to obtain a standard error of mean of 5% or less for all measured trait values. The correlation between falling velocity and other *H. sosnowskyi* propagule traits was measured using the North Group only. We assumed that the correlations between mericarp traits were specific to all representatives of the species and therefore, that it would be sufficient to make the detailed trait measurements and regression evaluation of trait relationships for one group only. The North Group propagules were collected at two sites 7.9 km apart. The two samples were combined into one according to the Kolmogorov–Smirnov test ($p$-value $= 0.97$ for propagule mass, $p$-value $= 0.61$ for propagule area). The median mass of *H. sosnowskyi* air-dry propagules of the North Group was $11 \pm 3$ mg; the median area of one propagule was $0.6 \pm 0.2$ cm$^2$; the median value of the wing loading was $0.019 \pm 0.005$ g / cm$^2$; and the median speed of its fall from a height of 2.7 m was $1.62 \pm 0.27$ m/s (Table 1).

The falling velocity depended on the propagule mass and propagule wing loading. The wing loading coefficient was the most important parameter of *H. sosnowskyi* propagules that affected their terminal velocity. Linear regression showed that this parameter was responsible for more than 80% of the terminal velocity variability. For a rough estimate of the terminal speed, one can use the propagule mass, the fluctuations of which are responsible for approximately 50% of the terminal velocity variability (Fig. 2).

To determine the relationship between the measured mean falling speed in still air and the release height, the propagules were dropped from heights varying from 0.80 to 4.28 m. Theoretically, as the release height increases, the measured fall rate should approach the terminal (constant) value. Two groups (A and B) of propagules that differed in *WL* by more than three times were selected for the measurements. The propagules of Group A had an abnormally low wing load: $N = 4$, $WL = 0.006 \pm 0.001$ g / cm$^2$, and 18 measurements of falling velocity. The propagules of Group B had a normal wing load: $N = 7$, $WL = 0.021 \pm 0.002$ g / cm$^2$, and 42 measurements of falling velocity. The results of linear regression showed that in the range of release heights used, it was not possible to determine the relationship between the measured mean fall rate and the release height (Fig. 3). Because the *H. sosnowskyi* propagules reached the terminal speed very quickly, we could adopt the

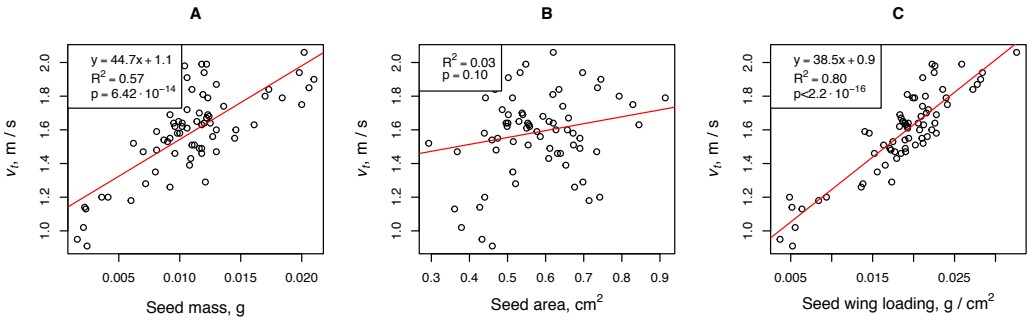

**Figure 2** **Relationships between *H. sosnowskyi* propagule terminal velocity and mass (A), area (B), and wing loading (C).** The terminal velocity was determined for a release height of 2.68 m. Linear regression is indicated by the red line.

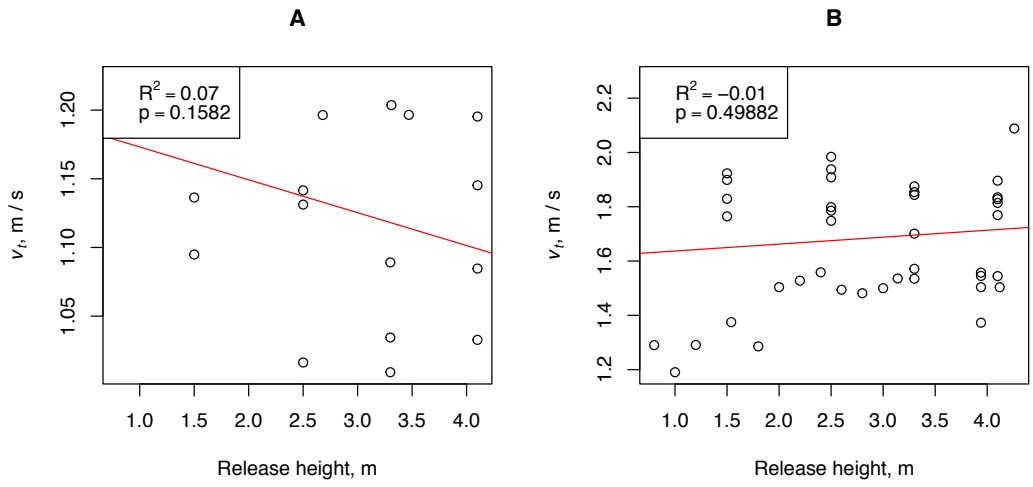

**Figure 3** **Relationship between *H. sosnowskyi* propagule terminal velocity and release height.** (A) propagules with a wing loading of $0.006 \pm 0.001$ g / cm², and (B) propagules with a wing loading of $0.021 \pm 0.002$ g / cm². Linear regression is indicated by the red line.

assumption of the simple ballistic model that the propagule reaches the terminal speed immediately after its release from the plant.

We tested the hypothesis that climate influences *H. sosnowskyi* propagule traits and release height (the height of umbels above ground level). We compared these characteristics for plants collected in two geographically distant regions: the city of Syktyvkar (North Group) and the city of Kazan (South Group) located approximately 6° latitude apart. Despite some differences between the traits of the North and South plant groups, their traits that directly affected the propagule flight distance can be considered almost identical, as shown by the Kruskal-Wallis test *p*-value $\geq 0.04$ (Fig. 4).

We performed 37 launches of paper propagule batches and 414 measurements of horizontal propagule flight distance (8–20 measurements per launch). The wind speeds

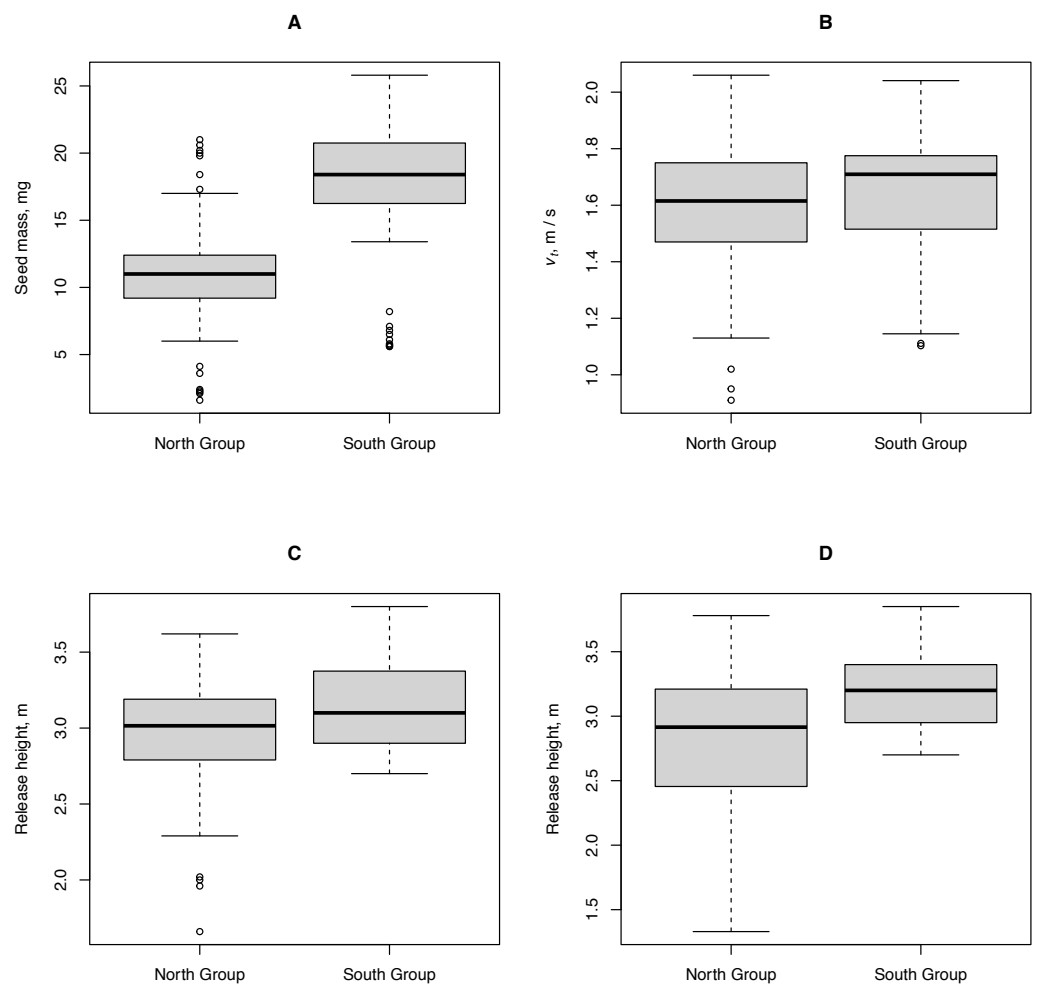

**Figure 4** **Characteristics of propagules and release heights of *H. sosnowskyi* from two geographically distant regions.** North Group: plants collected in the vicinity of Syktyvkar, and South Group: plants collected in the vicinity of Kazan. (A) Propagule mass, (B) terminal velocity, (C) height of the central umbels above ground level, and (D) height of the lateral umbels above ground level.

were between 0 and 9 m / s. Hereafter in the Results section, the word propagules refers to artificial propagule models made of paper, with standard shape and wing loading.

An initial analysis of the field measurements showed two key features of the relationships between wind speed and horizontal flight distance (Fig. 5). First, the flight distance of the propagules was strongly correlated with wind speed (Pearson correlation coefficient: 0.78, *p*-value $< 2.2 \cdot 10^{-16}$). Second, propagules of uniform shape, weight, and size, when dropped simultaneously from the same height, flew off and landed at different distances, and with increasing wind speed, the distance range increased. A positive significant correlation was observed between the values of the interquartile range of the flight distances of the standardized propagules and the wind speed (Pearson correlation coefficient: 0.74, *p*-value $< 1.4 \cdot 10^{-7}$). The results of modeling this relationship using linear regression are described below.

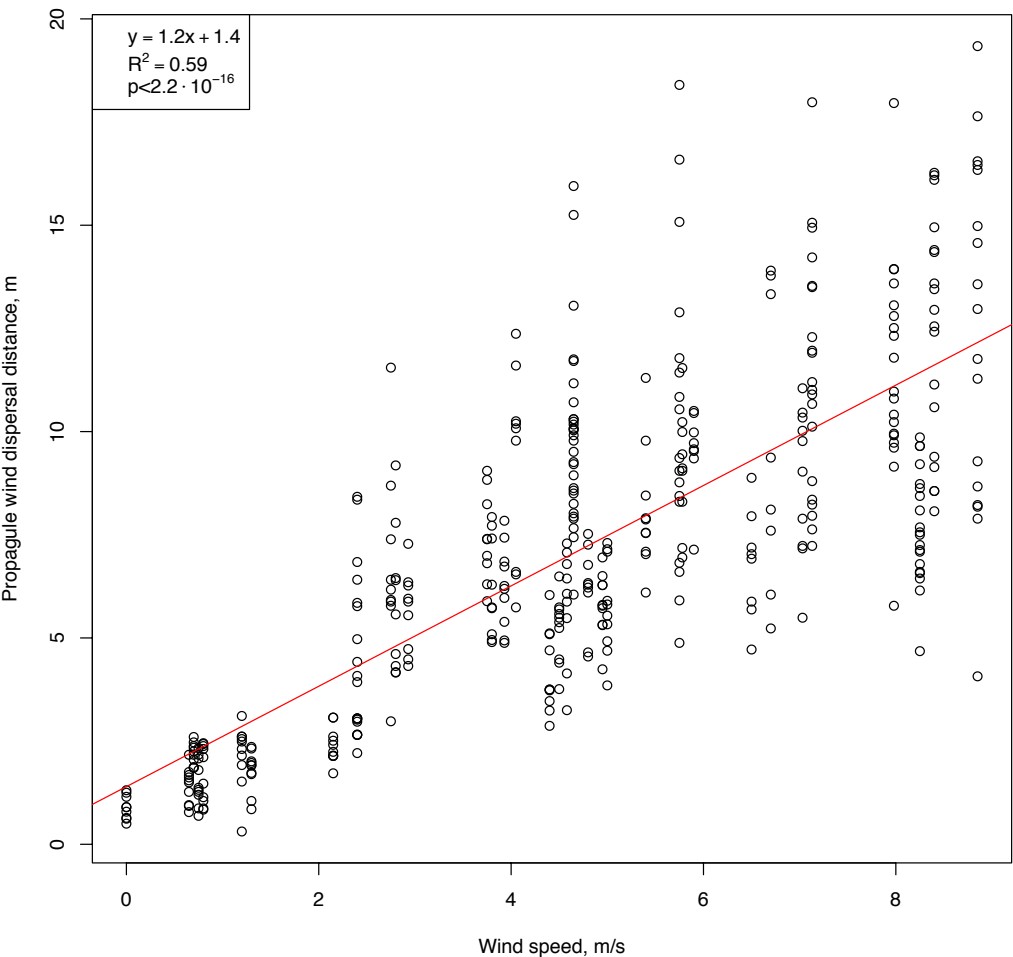

**Figure 5** **Raw results showing the flight distances of artificial *H. sosnowskyi* propagules at different wind speeds.** Linear regression is indicated by the red line.

We aggregated our empirical data for further analysis. We calculated the minimum ($D_{min}$), median ($D_{median}$), mean ($D_{mean}$), and maximum ($D_{max}$) propagule flight distances for each launch and correlated them with the aggregated wind speed measurements: median ($v_{median}$), average ($v_{mean}$), and maximum ($v_{max}$). The highest Pearson correlation coefficient (0.901, $p$-value $< 2.9 \cdot 10^{-14}$) was obtained for ($v_{max}$ and ($D_{mean}$). For further analysis, we used this pair of vectors. Linear regression of the horizontal flight distance ($D_{mean}$) on the wind speed ($v_{max}$) showed that more than 80% of the $D_{mean}$ variability was due to variability in $v_{max}$ (Fig. 6).

We calculated the theoretical horizontal flight distance of propagules using a simple ballistic model at the wind speeds ($v_{max}$) that we recorded during field experiments and compared them with empirical data using the Kolmogorov–Smirnov test and linear regression (Fig. 7A1, 7B1). The simple ballistic model appeared to be effective as the range of distances obtained from calculations using Eq. (1) did not differ significantly from the empirical data (Kolmogorov–Smirnov test: $p$-value $= 0.52$). This model enabled us to

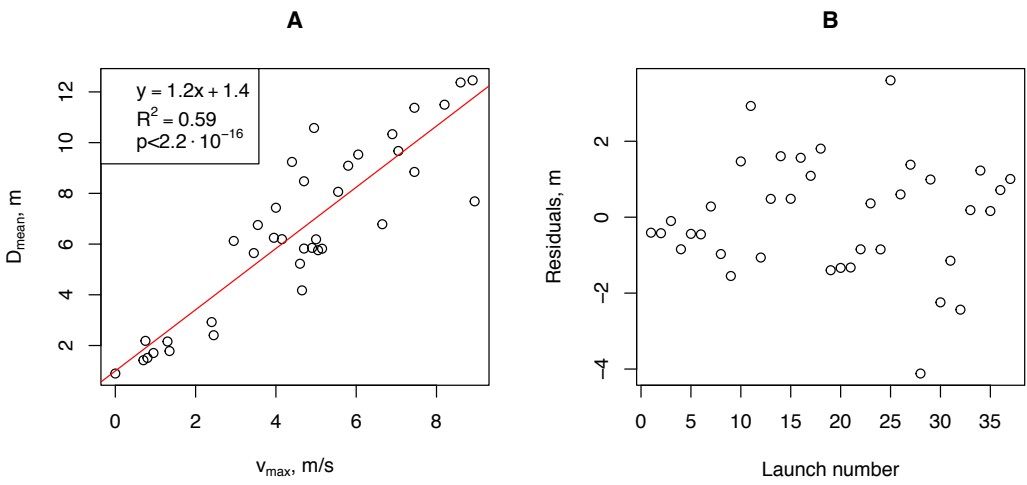

**Figure 6 The relationship between the artificial *H. sosnowskyi* propagules mean flight distance and maximum wind speed (A), and the regression residuals (B).** Linear regression is indicated by the red line.

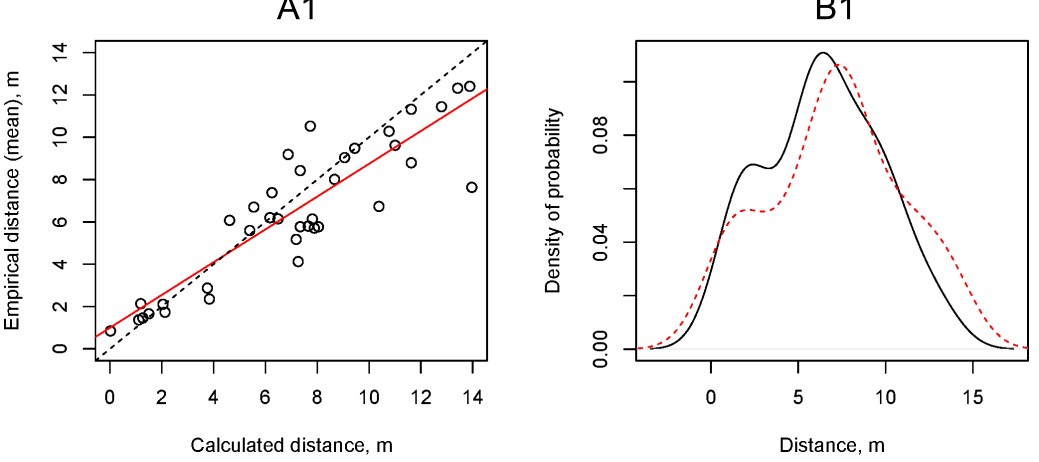

**Figure 7 Relationship between the artificial *H. sosnowskyi* propagules empirical flight distances and the flight distances calculated using the simple ballistic model (upper two plots) and the gradient model with $\alpha = 0.29$ (lower two plots).** (A) Results of linear regression (red line), dotted line is 1:1 line: (A1) $D_{mean} = 0.77 \pm 0.07$ (*p*-valu *e* < 0.001), Intercept = 1.00 ± 0.53 (*p*-value =0.07), (A2) $D_{mean} = 1.00 \pm 0.09$ (*p*-valu *e* < 0.001), Intercept = 1.00 ± 0.53 (*p*-value =0.07). (B) Kernel density estimation of the artificial *H. sosnowskyi* propagules empirical flight distances (black line) and that of simulated distances (red line). The results of the two-way Kolmogorov–Smirnov test showed that the two samples, in both cases, belong to the same statistical population: (B1) *p*-value =0.52, (B2) *p*-value =0.35.

explain 80% of the variability in the experimental $D_{mean}$ values. A significant shortcoming of the simple ballistic model was that the angle of the regression line reflecting the relationship between the calculated and experimental data differed from 45% (tangent = 1) and its tangent was equal to $0.77 \pm 0.07$.

The gradient model allowed us to consider the vertical wind speed fluctuations by using the ground surface roughness Hellmann exponent ($\alpha$). We used the same empirical data on wind speed ($v_{max}$) as for the simulation of propagule flight distance using the simple ballistic model. To find the coefficient $\alpha$ at which the regression line angle tangent between the calculated and empirical distances is equal to 1.00, we performed a series of the propagule flight distance calculations using the gradient model and changing the coefficient $\alpha$ from 0.05 to 0.50 with increments of 0.01. The value of $\alpha = 0.29$ provided the optimal convergence of the calculated and empirical data (Figs. 7A2, 7B2).

As shown in Fig. 5, the empirical data demonstrated that the propagule flight distance variation increased as the wind speed increased. The range between the minimum and maximum flight distances of the artificial propagules during the same launch linearly depended on the wind speed and ranged from 0.75 m at zero wind speed to 5.27 m at a wind speed of 8–9 m / s. We divided the distance measurement results into classes according to wind speed ($v_{max}$) and discretized them with increments of 1 m, and then calculated the standard deviation of the propagule distance flight ($D_{sd}$) for each wind speed class. The results of linear regression of $D_{sd}$ on $v_{max}$ showed a strong relationship between these parameters ($R^2 = 0.82$, $p$-value $= 4.13 \cdot 10^{-7}$). The linear regression equation is as follows:

$$D_{sd} = 0.35 \pm 0.06v + 0.51 \pm 0.29 \qquad (7)$$

where $D_{sd}$ is the standard deviation of the flight distances of propagules released at wind speed $v$. It should be noted that there were only nine pairs of $v_{max}$ and $D_{sd}$ because of data discretization. The limited number of pairs did not allow for the calculation of a statistically significant intercept ($0.51 \pm 0.29$, $p$-value $= 0.07$) (see primary data and R-script at https://doi.org/10.5281/zenodo.3837647).

## DISCUSSION

The range dynamics of plant species depend on the growth rate and dispersal of the plant population, and on the availability of suitable habitats (*Higgins & Richardson, 1999*). All of these processes need to be modeled to predict plant species distribution under global changes caused by climate change and human activity. This prediction is especially important for invasive species that have negative impacts on humans and ecosystems. Long-distance dispersal events are rare but have a radical effect on the spread rate (*Higgins & Richardson, 1999*). Modeling of LDD with a phenomenological approach is difficult owing to the rarity of LDD events. Mechanistic models of dispersal by wind allow us to use a few easily measurable variables: wind statistics, seed release height, and seed terminal velocity (*Katul et al., 2005*).

Simple mechanistic models with three main independent variables demonstrated reasonable prognostic accuracy for horizontal flight distances of *H. sosnowskyi* propagules. The comparison results showed that making the simple ballistic model more complex by adding a new parameter $\alpha$ (Hellmann exponent) is adequate. This parameter allowed us to account for a decrease in the wind speed in the vertical direction. The tangent of the regression line angle between the calculated and empirical distances became equal to 1

with an optimal value of $\alpha = 0.29$. These Hellmann exponent values correspond well to the values recommended for unstable air above human inhabited areas ($\alpha = 0.27$) and neutral air above human inhabited areas ($\alpha = 0.34$) (*Cleveland & Morris, 2013*).

These results are comparable with the results of field validation of a more complex mechanistic model that has been used to predict tree seed dispersal by wind (*Nathan, Safriel & Noy-Meir, 2020*). The reasonable prognostic accuracy of our simple mechanistic model can be explained by the fact that the *H. sosnowskyi* stem, inflorescences, and propagules together make up a much simpler system compared to that of tree species. *H. sosnowskyi* has relatively lower release heights. Its seeds do not bear special structures for increasing air drag force, such as samaras or long hairs, and have relatively large terminal velocity values($>$1 m / s). These *H. sosnowskyi* traits allowed us to ignore wind turbulence, and the description of vertical wind velocity was limited by incorporating the Hellmann exponent into the model.

Despite the availability of well-elaborated mechanistic models for describing and predicting propagule dispersal by wind, to date they have not been used for *Heracleum* species. Most cases of mechanistic modeling of wind seed dispersal deal with tree species (*Nathan et al., 2011*). For most herb species, wind cannot serve as a LDD agent because of the relatively lower release heights of most species. However, giant hogweeds (*H. sosnowskyi*, *H. mantegazzianum*, and *H. persicum* Desf.) are outstanding herbs with diaspore release heights of 2.5–3 m and mechanistic modeling of diaspore wind dispersal for these species is reasonable.

The lack of mechanistic models for giant hogweed propagule dispersal by wind can be attributed to the difficulty of testing these models by empirical dispersal distance measurement. Hundreds of propagules with identical terminal velocities are needed to provide sufficient replication of the measurements, and the same propagules cannot be used for repeated measurements because the wing-like border of the propagule is very fragile and can be easily broken after the first flight distance measuring cycle. However, our study has shown that the use of artificial propagules with a standard shape, wing loading, and terminal velocity close to the corresponding median values of natural propagules enabled us to overcome these difficulties.

The idea of using artificial propagules for dispersal modeling with different dispersal agents is not new. Artificial fruits have been used for the quantitative assessment of several fruit characteristics and wind speed as factors for horizontal distance transfer of tropical tree fruits (*Augspurger & Franson, 1987*). Artificial fruits were used to study the influence of odor, color, and nutrient content on fruit predators and dispersers (*Wang & Chen, 2009*; *Oliveira Barcelos, Perônico & Eutrópio, 2012*). As mentioned in the Methods section, the artificial propagule surfaces are much smoother, which may influence the air drag force. It should be noted that the dispersal distances obtained with the proposed model may be shorter, but are no longer than those for natural propagules under the same conditions. Nevertheless, we supposed that the fuzziness of horizontal flight distances (Eq. (7)) was more significant than the difference in the air drag force between artificial and natural propagules.

The finding of a significant difference in the flight distance of the artificial propagules after a simultaneous drop from the same height under the same wind conditions was unexpected. These differences are probably due to the aerodynamic properties of *H. sosnowskyi* propagules. The nearly oval propagules of the species have a strongly flattened shape, they do not have special accessories for aerodynamic stability (e.g., wings or a pappus), and the center of mass is close to the geometrical center of the propagule. However, this fuzziness can be considered and introduced into our mechanistic gradient model using Eq. (7).

In the Introduction section, we mentioned several studies performed to assess the dispersal of mericarps of giant hogweeds by air currents (*Clegg & Grace, 1974*; *Müllerová et al., 2005*; *Moravcova et al., 2007*; *Jongejans, Skarpaas & Shea, 2008*; *Pergl et al., 2011*; *Wojewódzka et al., 2019*). There is agreement among these authors that most giant hogweed propagules fall within a radius of 5–10 m from the parent plant. It has been supposed that the main LDD agents for *Heracleum* species are streams and human activity. We tested this supposition using the model proposed in this study.

Using the gradient model that we developed and weather station data, we were able to assess the expected maximum *H. sosnowskyi* propagule dispersal distance. We used the maximum wind velocity recorded at the Syktyvkar and Kazan airport weather stations from 2013–2020 during August-October (the period of fruit readiness for dispersal before snow cover formation). The maximum wind velocity for Syktyvkar was 16 m / s and for Kazan, 23 m / s at a height of 10 m. Given that the maximum *H. sosnowskyi* release height registered in our study was 3.85 m, and taking into account the wind gradient, the corrected wind velocities for a height of 3.85 m were calculated as 12 m / s for Syktyvkar and 17 m/s for Kazan. Considering that the slowest terminal velocity, determined for real *H. sosnowskyi* propagules, was 0.91 m / s, the expected dispersal distance for such mericarps was calculated as $39 \pm 5$ m for Syktyvkar and $55 \pm 6$ m for Kazan. For propagules with a median terminal velocity of 1.65 m / s, the expected dispersal distance was calculated as $22 \pm 6$ m for Syktyvkar and $31 \pm 7$ m for Kazan. A quick exploration of wind gust statistics for Syktyvkar and Kazan showed that the Kazan climate is significantly windier, and the minimum, median, and maximum wind gust velocities were approximately 30% higher in Kazan than in Syktyvkar.

These draft calculations show that wind should not be excluded from the list of LDD agents for *Heracleum* species. The proposed mechanistic model can explain published observations on the invasion spread rate (up to 26.7/year) of *H. mantegazzianum* (*Müllerová et al., 2005*) as well as individual observations of the extreme flight distances of *H. sosnowsky* propagules of up to 50 m (*Ochsmann, 2008*).

The traits and release heights of *H. sosnowskyi* propagules are relatively uniform over large geographical areas with different climates. The observed differences in terminal velocities and release heights were not sufficient to significantly affect the horizontal flight range of the propagules. The climate significantly affects only the timing of fruit ripening and wind conditions. A dataset with *H. sosnowskyi* propagule traits (measurements for 130 propagules) and release heights (290 measurements) published in Zenodo (https://doi.org/10.5281/zenodo.3837647) enabled us to model the propagule dispersal

distances by wind over the entire invasion range of this species. The only additional data needed for modeling of propagule dispersal by wind in a specific part of the invasion range was data from the closest weather station.

Within one population, *H. sosnowskyi* propagules may have significant differences in terminal velocity and may be located at different heights above ground level. These differences can lead to significant differences in propagule dispersal by airflow. Therefore, to calculate the entire range of distances of propagule dispersal using the proposed mechanistic gradient model, it is necessary to develop an individual-based model (IBM), in which the flight of each propagule is calculated separately. The values of the *H. sosnowskyi* propagule terminal velocity and release heights should be selected randomly from a variety of empirically obtained data. Wind speeds for a specific area should be available for the period from the beginning of propagule maturation until all the propagules capable of releasing have been dispersed.

Our observations showed that the propagules of *H. sosnowskyi* have different release capacities from the inflorescence: some propagules fall off at the slightest vibration of the plant shoot at almost zero wind speed, whereas some remain on the umbels even after strong gusts of wind exceeding 15 m/s. The dragging of individual propagules or umbrellas with the remaining propagules over the surface of snow by wind requires additional studies. To model the flight distances of propagules released from umbels, it is important to consider that different groups of propagules in the same umbel may have their own critical wind speeds. The critical wind speed of a propagule is the minimum wind speed at which it is released from the umbel.

An important requirement for obtaining adequate results using an IBM is the availability of high-quality measurements of the wind speed for a given area at the highest possible frequency. The most appropriate information for our purposes was provided by the airport weather stations. The weather stations measure the wind speed every 30 min, and information about the maximum wind gusts between measurement periods was also available.

## CONCLUSION

Our findings showed that the wind contribution to propagule dispersal of invasive *Heracleum* species has been strongly underestimated in most studies. However, more detailed and accurate results will be available after application of the IBM that we developed. This will enable us to calculate the flight distances of *H. sosnowskyi* propagules, taking into account real weather conditions during different years in different parts of the invasion range of this species. We will be able to describe the direction and dynamics of *H. sosnowskyi* expansion in unoccupied territories and provide practical recommendations for the management of this invasive species.

### Funding

The reported study was funded by RFBR and NSFB, project number 20-54-18002 and was partially supported within the scope of State Tasks for IB FRC Komi SC UB RAS (GR no. AAAA-A17-117033010038-7). The funders had no role in study design, data collection and analysis, decision to publish, or preparation of the manuscript.

### Grant Disclosures

The following grant information was disclosed by the authors:
RFBR and NSFB: 20-54-18002.
State Tasks for IB FRC Komi SC UB RAS: GR no. AAAA-A17-117033010038-7.

### Competing Interests

The authors declare there are no competing interests.

### Author Contributions

- Ivan Chadin and Igor Dalke conceived and designed the experiments, performed the experiments, analyzed the data, prepared figures and/or tables, authored or reviewed drafts of the paper, and approved the final draft.
- Denis Tishin and Ruslan Malyshev conceived and designed the experiments, performed the experiments, authored or reviewed drafts of the paper, and approved the final draft.
- Ilya Zakhozhiy conceived and designed the experiments, performed the experiments, analyzed the data, authored or reviewed drafts of the paper, and approved the final draft.

### Data Availability

The primary data and the R-scripts are available at Zenodo: Chadin Ivan, Dalke Igor, Tishin Denis, Zakhozhiy Ilya, & Malyshev Ruslan. (2020). Dataset and R-script for simple mechanistic model of *Heracleum sosnowskyi* seed dispersal by wind (Version 1.0) [Data set]. Zenodo. http://doi.org/10.5281/zenodo.3837647.

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
