# Peer review of "A simple mechanistic model of the invasive species Heracleum sosnowskyi propagule dispersal by wind"

_PeerJ, doi:10.7717/peerj.11821_

## Round 0.1 · original submission · Major Revisions

Dear Authors,

We now received two thorough reviews for your manuscript. Both reviewers agree that this is an interesting manuscript reporting on a well-performed study, but both also point to a number of issues that will need to be adressed.

A major focus for assessing the revised version will be the discussion section: It is of very high importance that you set your own results into context. Please thoroughly search for existing, relevant studies and relate your results to what they have found (see comments on this by Reviewer #2). Another important aspect is to make sure the range of applicability of your findings is clear - see the remarks of Reviewer #1 concerning the taxonomic focus of your study.

When revising your manuscript, please make sure to consider all suggestions made by the reviewers, also those included in the pdf provided by Reviewer #2, and provide a point by point response.

I am looking forward to your revision.
Kind regards,
Tina Heger

·

Basic reporting

1. Basic Reporting
• Clear and unambiguous, professional English used throughout.
o OK. Very few linguistic aberrations.
• The article must be written in English and must use clear, unambiguous, technically correct text. The article must conform to professional standards of courtesy and expression.
o Generally OK.
o One significant detail: technically the Apiacae do not disperse ”seeds”. Rather, the dispersed unit is in fact part of a cypsela (A fruit similar to an *achene except that it develops from an inferior ovary and thus also includes noncarpellary tissue; http://www.botanydictionary.org/cypsela.html). Apiacaean propagules are normally developed in pairs (each umbel carryijng on pair) that split up before dispersal, each part disseminating individually. To avoid confusion it might be wise to throughout use e.g. ”propagule” instead of ”seed”. To add a bit to the framework of the potential consequences of Heracleum propagule dispersal it may be worth conidering whether the invasivity is affected since as in Heracleum spp, each dispersed propagule does not contain a mature seed immediately ready to germinate, but a fertilized ovule that will need to mature lying in the ground before being able to germinate (if not consumed meanwhile, e.g.).
o The title refers to conclusions regarding the genus Heracleum but the investigation covers dispersal of H. sosnowskyi propagules (and artifical disseminules representing H. sosnowskyi. I suggest adding ”sosnowskyi” wherever appropriate will clarify this.
• Literature references, sufficient field background/context provided.
o OK.
• The article should include sufficient introduction and background to demonstrate how the work fits into the broader field of knowledge. Relevant prior literature should be appropriately referenced.
o OK.
o Comment: The reason for the investigation is to add to the background in regard to the invasiveness of H. sosnowskyi (or, as commented above the genus Heracleum, which is not fully correct). Admittedly H. sosnowskyi is regarded as invasive i central and eastern Europe, just like certain other Heracleum species (e.g. H. mantegazzianum) whereas others are not (H. sphondylium).
• Professional article structure, figures, tables. Raw data shared.
o OK
• The structure of the article should conform to an acceptable format of ‘standard sections’ (see our Instructions for Authors for our suggested format). Significant departures in structure should be made only if they significantly improve clarity or conform to a discipline-specific custom.
o OK
• Figures should be relevant to the content of the article, of sufficient resolution, and appropriately described and labeled.
o OK
Comment: Contents of lines 178-184 is repeated in Table 1. The latter is not absolutely necessary (but OK).
o Comment: Likewise Fig. 4, unless interesting to demonstrate the actual propagule mass, wing loading and height of release, illustrating non-significant differences (except in the case of sed mass differences) is slightly superfluous (again OK, space permitting publication).
o Comment: Fig. 5 ”Seed fly distance” should rather read ” Propagule wind dispersal distance”.
o Comment: Caption of Figs. 6 and 7 should include ”artificial H. sosnowskyi propagules”.
• All appropriate raw data have been made available in accordance with our Data Sharing policy.
o OK – I believe…could not open, lacking appropriate app. (and time to download one). Sorry!
• Self-contained with relevant results to hypotheses.
o OK
• The submission should be ‘self-contained,’ should represent an appropriate ‘unit of publication’, and should include all results relevant to the hypothesis.
o OK
• Coherent bodies of work should not be inappropriately subdivided merely to increase publication count.
o OK

Experimental design

2. Experimental design
• Original primary research within Aims and Scope of the journal.
o OK
• Research question well defined, relevant & meaningful. It is stated how research fills an identified knowledge gap.
o OK.
o Comment: interesting topic and result; needs further investigation adding other factors affecting the success of H. sosnowskyi invading ”new” regions (moving frontline (e.g. meters/year, germinability - %) and the reason for this. But OK, this a topic for discussion. Prpopagule dispersal alone is not enough for successful invasion.
• The submission should clearly define the research question, which must be relevant and meaningful. The knowledge gap being investigated should be identified, and statements should be made as to how the study contributes to filling that gap.
o OK. See above.
• Rigorous investigation performed to a high technical & ethical standard.
o OK
• The investigation must have been conducted rigorously and to a high technical standard. The research must have been conducted in conformity with the prevailing ethical standards in the field.
o OK
• Methods described with sufficient detail & information to replicate.
o OK
• Methods should be described with sufficient information to be reproducible by another investigator.
o OK

Validity of the findings

3. Validity of the findings
• Impact and novelty not assessed. Negative/inconclusive results accepted. Meaningful replication encouraged where rationale & benefit to literature is clearly stated.
o OK
• All underlying data have been provided; they are robust, statistically sound, & controlled.
o OK. (But I have not performed repeated staistical analysis.)
• The data on which the conclusions are based must be provided or made available in an acceptable discipline-specific repository. The data should be robust, statistically sound, and controlled.
o OK. As above: and as described above I could not retrieve raw data.
• Conclusions are well stated, linked to original research question & limited to supporting results.
o OK
• The conclusions should be appropriately stated, should be connected to the original question investigated, and should be limited to those supported by the results. In particular, claims of a causative relationship should be supported by a well-controlled experimental intervention. Correlation is not causation.
o OK. Comments above relevant here.
• Speculation is welcome, but should be identified as such.
o OK

Additional comments

4. General comments
• Being interestad in this topic I find this paper welcome and filling a knowledge gap.
• Well performed and described empirical investigation and experiment – still more components affecting dispersal need attention.
• It would be interesting to know whether solitary plants are as efficient as larger monospecific stands of these plants dispersing into new lands (´guerilla´vs phalanx´ strategy?).
• The aerodynamics of disk-shaped propagules with centrally positioned point of gravity is likely to be strongly affectd by the angle of incidence as the wind cathches it, and likewise knowing what force is needed to disassemble the cypsela from the umbel would add to the understanding of the dispersal distance (differs with maturity of propagule).
• Conclusions imply that further investigation is under way and will reveal more information about the westward spread of H. sosnowskyi (and likely other Heracleum species and other wind dispersed plants as well).

·

Basic reporting

The reviewed manuscript is written clearly. There are only a few parts that need some clarification. I made my comments directly to the attached file. The structure of the paper is relatively good. I see a problem in the discussion, where authors discuss only their results. I need some more text devoted to putting the work in the context of previous works etc.

Experimental design

The experimental design is clearly described. I miss a wider description/discussion of whether the used paper seeds can give different results than natural seeds.

Validity of the findings

The results are clear and are linked to the original research question. The only comment goes to the fact, that only observations in one region were taken. It should be clearly explained.

Additional comments

I made my comments directly to the review file. The major objection goes to the weird discussion which does not include references. Some parts of the introduction need to put your work to the wider context.
The design is described clearly with sound results.

---

## Round 0.2 · Minor Revisions

Dear Authors,

Thank you for addressing the reviewer comments so carefully. Reviewer #2, who had some serious points of critique in the first round, was unfortunately not available for reviewing the revision. But judging from my own assessment, his comments have been addressed well.

However, I would like to ask you to carefully check this new version for grammar and other language issues, since it contains quite a number of mistakes, especially in the newly added sections.

Here are some examples:
Abstract: "Dispersal of propagules is the first stage of
plant species spreading and strongly influences [instead of influence on] the dynamics of biological invasion."
"Heracleum sosnowskyi is one of the most dangerous invasive [instead of invasion] species"
"We measured and compared the propagule traits" [instead of propagules traits - also in other instances in the text]

Beginning of the discussion section:
Instead of "The plant species range dynamics depends on the plant populations growth rate, dispersal, and availability of suitable habitats" I suggest "Range dynamics of plant species depend on ..."; "depends" is a grammar mistake.
Line 276: 'is' is missing

These are just a few examples - please make sure that the second revision does not contain any grammar or other mistakes, and consider getting help from a fluent English speaker.

Kind regards,
Tina Heger

---

## Round 0.3 · accepted · Accept

Dear Authors,

Thank you for having your manuscript checked by a professional editing service. This new version has much improved in language, and in my point of view, it is now ready for publication in PeerJ.

Kind regards,
Tina Heger